# Hereditary Ataxia: A Focus on Heme Metabolism and Fe-S Cluster Biogenesis

**DOI:** 10.3390/ijms21113760

**Published:** 2020-05-26

**Authors:** Deborah Chiabrando, Francesca Bertino, Emanuela Tolosano

**Affiliations:** Molecular Biotechnology Center (MBC), Department of Molecular Biotechnology and Health Sciences, University of Torino, 10126 Torino, Italy; francesca.bertino721@edu.unito.it (F.B.); emanuela.tolosano@unito.it (E.T.)

**Keywords:** ataxia, FLVCR1, FRATAXIN, ABCB7, PCARP, FRDA and XLSA/A

## Abstract

Heme and Fe-S clusters regulate a plethora of essential biological processes ranging from cellular respiration and cell metabolism to the maintenance of genome integrity. Mutations in genes involved in heme metabolism and Fe-S cluster biogenesis cause different forms of ataxia, like posterior column ataxia and retinitis pigmentosa (PCARP), Friedreich’s ataxia (FRDA) and X-linked sideroblastic anemia with ataxia (XLSA/A). Despite great efforts in the elucidation of the molecular pathogenesis of these disorders several important questions still remain to be addressed. Starting with an overview of the biology of heme metabolism and Fe-S cluster biogenesis, the review discusses recent progress in the understanding of the molecular pathogenesis of PCARP, FRDA and XLSA/A, and highlights future line of research in the field. A better comprehension of the mechanisms leading to the degeneration of neural circuity responsible for balance and coordinated movement will be crucial for the therapeutic management of these patients.

## 1. Introduction

Balance and coordinated movements of the body are the result of a complex neuronal network involving the basal ganglia, cerebellum and cerebral cortex as well as peripheral motor and sensory pathways. Alterations in any part of this circuitry lead to loss of coordination and balance, also termed ataxia [1]. The etiology of ataxias is complex and includes toxic, metabolic, immune and genetic causes [2,3,4,5]. Hereditary forms represent an important subgroup of the ataxic disorders with more than 70 different types described [1,6,7]. Unfortunately, only few types of ataxia are fully treatable, while the majority are still only symptomatically managed, mostly because the molecular pathogenesis of the disease is poorly understood [1]. Hereditary ataxias can be divided by mode of inheritance in autosomal dominant and autosomal recessive disorders, and by gene in which causative mutations occur [7]. A wide range of mutations have been described in different genes resulting in the alteration of distinct mechanisms as mitochondrial function, oxidative stress, DNA repair, protein folding, cytoskeletal proteins, heme metabolism and iron–sulfur [Fe-S] cluster biogenesis [1].

The present review focuses exclusively on those forms of hereditary ataxias caused by mutations in genes involved in heme metabolism or Fe-S cluster biogenesis (Table 1), the two major iron consuming processes in the cell. Despite recent progresses in elucidating the role of Fe-S clusters in the maintenance of the neuronal networks at the basis of coordination and balance, less is known about the role of heme in these processes. Overall, the pathological mechanisms at the basis of neurodegeneration in these forms of ataxia are still unclear. The aim of this review is to provide a detailed description of the biology of heme metabolism and Fe-S cluster biogenesis with a focus on the possible altered molecular mechanisms at the basis of ataxia. 

## 2. Heme Metabolism and Fe-S Cluster Biogenesis

Heme and Fe-S clusters are iron-containing cofactors involved in a variety of processes indispensable for cell viability. Heme, a complex of iron and protoporphyrin IX, is required for oxygen transport, cell respiration, drug metabolism, ion channel function and gene regulation [15,16,17,18,19,20,21]. Fe-S clusters are a complex between iron and the inorganic sulfurs. The most frequent forms include [2Fe-2S], [3Fe-4S] and [4Fe-4S]. However, more complicated forms have been characterized [22]. Fe-S clusters are involved in cell respiration, cell metabolism, genome maintenance, ribosome biogenesis and antiviral defense [23,24,25,26].

Both heme and Fe-S clusters are endogenously synthesized through complex and highly regulated processes occurring between the mitochondria and cytosol. The scenario is complicated by the requirement for heme and Fe–S clusters in distinct subcellular compartments coupled to their hydrophobicity and potential cytotoxicity. Therefore, the intracellular delivery of these cofactors to target apoproteins requires the binding of heme and Fe-S clusters to specific cochaperone and chaperones systems [18,19,20,21,23,24,25,26]. All these fascinating mechanisms have been just recently elucidated, but several questions still remain unanswered.

The heme biosynthetic pathway involves eight enzymatic reactions that are compartmentalized between the mitochondrion and cytosol. Heme synthesis starts and terminates in the mitochondrial matrix but the intermediate enzymatic steps occur in the cytosol or in the intermembrane space of mitochondria, meaning that heme precursors must cross mitochondrial membranes. 

The initial step of heme biosynthesis is the condensation of succinyl-CoA and glycine, by δ-aminolevulinic acid synthase (ALAS), to form δ-aminolevulinic acid (ALA) in the mitochondrial matrix. Notably two different isoforms of ALAS exists: the ubiquitously expressed ALAS1 and the erythroid-specific isoform ALAS2. Upon its synthesis, ALA is exported into the cytosol where it is converted into coproporphyrinogen III (CPgenIII) through four subsequent enzymatic reactions. CPgenIII is then translocated in the intermembrane space of the mitochondria where it is converted into protoporphyrin IX (PPIX) in two enzymatic steps. A detailed description of each of these reactions and the putative transporters involved in the translocation of heme precursors through mitochondrial membranes can be found elsewhere [17,27]. In the last step, ferrous iron is inserted into the protoporphyrin ring by ferrochelatase (FECH) to form heme (heme *b*). Iron delivery to mitochondria is ensured by the iron importer mitoferrin (MFRN) located in the inner mitochondrial membrane [28,29]. Independent studies showed that FECH interacts with other proteins involved in heme synthesis—such as MFRN [30], the ATP binding cassette proteins ABCB7 [31,32] and ABCB10 [30,32], ALAS2 [33], and enzymes involved in the succinyl-CoA production [34,35]—thus supporting the existence of a large complex mediating heme synthesis, also termed mitochondrial heme metabolism complex [33]. 

Upon its synthesis heme b and/or its derivatives heme *a*, heme *o*, and heme *c* are incorporated into target apoproteins that reside in different subcellular compartments (mitochondria, ER, golgi, nucleus, lysosomes and peroxisomes) [20]. Therefore, once synthesized, heme must first cross the two mitochondrial membranes and then reach specific organelles to be incorporated into apoproteins. How this is achieved is still a matter of debate. The mitochondrial isoform of feline leukemia virus subgroup C receptor 1 (FLVCR1), FLVCR1b, plays a role in the efflux of heme out of mitochondria [36]. However, alternative potential mechanisms of mitochondrial heme export have been proposed [17]. Due to the hydrophobic nature and cytotoxic properties of free-heme, the intracellular trafficking of heme to target subcellular compartments must be finely controlled as well [18,21,37]. This is likely achieved by heme binding to specific chaperones [38,39]. 

The bioavailability of free-heme is further controlled by heme catabolism, mediated by heme oxygenase 1 (HO1) [40], and heme trafficking across the plasma membrane. Notably, heme transporter HRG1 [41], heme transporter HRG4 [42] and feline leukemia Virus subgroup C receptor 2 (FLVCR2) [43] have been described as heme importers whereas FLVCR1 [44], the multidrug resistance protein MRP-5/ABCC5 [45] and the ATP binding cassette subfamily G member 2 (ABCG2) [46,47] as heme exporters. Although the physiological significance of heme export out of the cell is still not completely understood, several lines of evidence suggest that heme export out of the plasma membrane is coupled with heme synthesis. Indeed, the export of heme through the plasma membrane isoform of FLVCR1, FLVCR1a, was reported to be essential for the regulation of ALAS1 activity in hepatocytes [48]. Consistently, the transcription of *Flvcr1a* and *Alas1* are upregulated upon the induction of heme synthesis [48]. Altogether, these data suggest that heme export is something more than a simple cell detoxifying mechanism, as initially proposed. Since heme inhibits its own synthesis by negatively regulating *ALAS1* transcription, translation and stability [16], it is tempting to speculate that heme export out of the plasma membrane is necessary to sustain endogenous heme synthesis.

The biogenesis of Fe-S clusters and their insertion into apoproteins is another extremely complex process involving more than 30 known biogenesis factors located in mitochondria and cytosol. The mitochondrial iron-sulfur cluster assembly (ISC) machinery is responsible for the de novo synthesis of Fe-S clusters and their insertion into mitochondrial apoproteins. The first step involves the assembly of a [2Fe-2S] cluster on the scaffold protein ISCU2 from inorganic iron and sulfur. The sulfur is donated by a cysteine desulfurase complex named NFS1/ISD11/ACP1 [49]. The mechanism of iron insertion into the cluster has been long debated. Initial studies suggest FRATAXIN (FXN) as an iron donor for [2Fe-2S] clusters assembly [50,51,52]. However, subsequent investigations propose FXN as an allosteric regulator of the process, able to stimulate the rate of sulfur transfer to the nascent Fe-S cluster [53,54,55,56,57,58]. Nowadays, the preloading of ISCU2 with iron has been proposed as major mechanism [23]. 

Following the synthesis of the [2Fe-2S] cluster on the ISCU2 scaffold, the [2Fe-2S] cluster is delivered to glutaredoxin 5 (GLX5) [23,59] to be incorporated into target [2Fe-2S] proteins. Alternatively, the [2Fe-2S] cluster is converted into [4Fe-4S] cluster and incorporated into different mitochondrial apoproteins, including aconitase, lipoyl synthase and respiratory cytochromes [23].

The ISC machinery also plays an essential role in the synthesis of a sulfur-containing compound (X-S) that is exported from mitochondria by the inner membrane ABC transporter, ABCB7 [60,61].

In the cytosol, X-S is used to generate the Fe/S clusters for cytosolic and nuclear target proteins as well as for those associated to the endoplasmic reticulum. These steps are mediated by the cytosolic iron-sulfur protein assembly (CIA) machinery that comprises at least 11 proteins [23,24]. For additional details on the molecular mechanisms underlying Fe-S clusters biogenesis, the reader is referred to additional outstanding reviews [23,24,25,26].

Several lines of evidence indicate that heme metabolism and Fe-S cluster biogenesis are strongly interconnected (Figure 1). Interestingly, heme synthesis depends on Fe-S cluster biogenesis. As mentioned above, the synthesis of heme requires succinyl-CoA as a first substrate. Succinyl-CoA derives from the TCA cycle that is also dependent of Fe-S cluster proteins. Notably, the supply of acetyl-CoA to the tricarboxylic acid cycle (TCA) cycle is mediated by the pyruvate dehydrogenase complex (PDC), that converts pyruvate into acetyl-CoA. The synthesis of PDC requires lipoate that is generated by an Fe-S cluster enzyme, the lipoyl synthase (LIAS) [62]. Furthermore, two TCA cycle enzymes, succinate dehydrogenase B (SDHB) and aconitase 2 (ACO2) are Fe-S cluster proteins [63,64].

The condensation of succinyl-CoA and glycine by ALAS represents the rate limiting step in heme biosynthesis and, in erythroid cells, it is dependent on Fe-S clusters levels. The cytosolic Fe-S cluster protein aconitase is a bifunctional enzyme and, following the loss of its Fe-S cluster (in iron deficiency), functions as an iron responsive protein (IRP) [64,65] inhibiting the translation of ALAS2 [66,67,68]. Moreover, GRX5 deficiency reduces heme synthesis in human erythroblasts by affecting Fe-S cluster biogenesis and IRPs activity [69]. 

In every cell type, the last step of heme synthesis is dependent on Fe-S cluster biogenesis. Notably, FECH contains an [2Fe-2S] cluster that does not participate in catalysis [70,71,72]. Reduced maturation and stabilization of FECH was observed in conditions of decreased availability of Fe-S clusters, like in ISCU miopathy [73], frataxin deficiency [74] or sideroflexin 4 deficiency [75].

Furthermore, as previously mentioned, FECH is part of a large complex of proteins, including the Fe-S cluster protein ABCB7 [32]. The downmodulation of ABCB7 results in low Fe-S clusters availability, decreased FECH expression and reduced heme levels [32].

Collectively, these observations suggest that the alteration of Fe-S cluster biogenesis may impact on heme levels by affecting multiple steps of its biosynthetic pathway, the supply of succinyl-CoA substrate as well as the expression of the first and last enzymes involved in heme synthesis. Whether alteration of heme biosynthesis affects Fe-S cluster biogenesis is not clear. 

Since heme synthesis and Fe-S clusters biogenesis represent the two major Fe consuming processes in mitochondria, the two pathways may compete for iron acquisition. Studies in differentiating erythroid cells suggested that FXN may act as a metabolic switch in the diversion of iron between Fe-S cluster biogenesis and heme synthesis. Indeed, protoporphyrin IX (PPIX) accumulation results in downregulation of FXN and in the redirection of iron from Fe-S clusters biogenesis toward heme biosynthesis [76].

Finally, it is interesting to note that heme and Fe-S clusters are both necessary for oxidative phosphorylation. Indeed, heme is a cofactor for cytochromes c and cytochromes in complexes II-III-IV of the mitochondrial electron transport chain (ETC) [77,78], whereas Fe-S clusters are cofactors for cytochromes in complexes I-III [23]. The observation that both heme and Fe-S clusters are required for ETC further suggests that the two pathways must be tightly co-regulated to ensure energy production.

## 3. Posterior Column Ataxia and Retinitis Pigmentosa (PCARP)

PCARP is a rare autosomal recessive neurodegenerative syndrome characterized by sensory ataxia and retinitis pigmentosa [8]. The disease begins in infancy with areflexia and signs of retinal degeneration that worsen over time with a progressive restriction of the visual field and loss of the retina function. Ataxia is clinically evident in the second decade of life. It is due to the progressive degeneration of the posterior columns of the spinal cord resulting in the loss of proprioceptive sensations. Affected individuals have no clinical or radiological evidence of cerebral or cerebellar involvement. Additional features of the disease include scoliosis, camptodactyly, achalasia and gastrointestinal dysmotility [9,79].

The disease-causing gene was first mapped to chromosome 1q31-q32 [8] and then identified as *FLVCR1* [79]. *FLVCR1* gene codes for two heme export proteins: the plasma membrane isoform FLVCR1a and the mitochondrial isoform FLVCR1b [36]. Both homozygous missense mutations and compound heterozygous mutations have been described in several families [79,80,81,82,83]. Interestingly, mutations in the same gene have been identified in patients with non-syndromic retinitis pigmentosa [84,85,86,87,88] or in patients with pain insensitivity [89,90,91]. All the *FLVCR1* mutations identified to date in PCARP patients are reported in Table 2.

The majority of mutations identified so far in PCARP occur in the first exon of the *FLVCR1* gene (Table 2) [36]. As FLVCR1b protein arises from an alternative transcription start site located in the second exon of the *FLVCR1* gene, the majority of identified mutations likely affect only FLVCR1a. However, the impact of the mutations on the expression and function of the two FLVCR1 isoforms still remains to be investigated in detail. The observation that loss of *Flvcr1* is incompatible with life in animal models [36,92,93] strongly suggests that the identified mutations in *FLVCR1* did not completely abrogate the expression of the transporter. This statement is also supported by data obtained in cells expressing the *FLVCR1* mutations identified in patients with pain insensitivity [89,90].

The pathophysiology of the disease is still completely unknown. FLVCR1 is a ubiquitously expressed heme exporter belonging to the major facilitator superfamily (MFS) of transporter [44,94,95]. Initial studies focused on the role of FLVCR1 isoforms during erythroid differentiation [36,92,93] and erythroid diseases [96,97]. Then, heme efflux through FLVCR1a has been involved in the regulation of hepatic P450 cytochromes [48], the maintenance of endothelial integrity [98] as well as the regulation of cell proliferation [99,100]. These data are consistent with a role for heme in multiple essential biological processes. Curiously, almost no information is available regarding the function of FLVCR1 in the nervous system. The reported animal models of FLVCR1 deficiency are not viable thus limiting the possibility to investigate the role of FLVCR1 in neurodegeneration, specifically in the retina and in the peripheral nervous system. Information concerning the disease pathogenesis now available have been obtained in cell lines overexpressing the pathogenetic mutations. It has been shown that *FLVCR1* mutations cause the mislocalization of FLVCR1 protein and consequently loss of its heme export function [101]. Therefore, it has been proposed that loss of FLVCR1 function may cause heme accumulation and cellular damage by heme toxicity in the cells of PCARP patients. Increased oxidative stress and enhanced susceptibility to programmed cell death have been observed also in neuroblastoma cells upon FLVCR1a downmodulation [89]. However, these experiments are not exhaustive and studies in appropriate in vitro and in vivo models of the disease are absolutely required. As discussed in the previous paragraph, several data support the idea that the export of heme is co-regulated with heme synthesis, thus suggesting that defective heme export may lead to decreased heme synthesis. Furthermore, defective heme export may also induce different compensatory mechanisms including the induction of HO1 with the release of carbon monoxide, iron and bilirubin, that could compromise the function and survival of proprioceptive neurons. Furthermore, the alteration of heme metabolism may affect additional pathways—including proteostasis, energetic metabolism and/or ion channel function [102]—leading to neurodegeneration. Future research should be focused on the generation of good in vitro and in vivo models of the diseases in order to definitely understand the consequences of defective heme export and its contribution to the disease pathogenesis. 

## 4. Friedreich’s Ataxia (FRDA) 

Friedreich’s ataxia (FRDA) is the most common form of recessive inherited ataxia in the Caucasian population, accounting for 75% of ataxias with onset prior to 25 years of age [103]. Symptoms usually manifest during childhood, and in 10 years patients lose the ability to walk, stand or sit up unassisted [10,11]. To note, in a small portion of cases, FRDA develops later on and it is characterized by a milder phenotype and a slower disease progression [12]. FRDA patients present a complex set of clinical features throughout the disease, including a mixed spinocerebellar and sensory ataxia, dysarthria, muscular weakness and absence of tendon reflexes with deep sensory loss [104].These neurological signs are the results of pathological changes that disturb both the central and peripheral nervous systems. Neurodegeneration firstly occurs in dorsal root ganglia (DRG) with progressive loss of large sensory neurons and their axonal projection in the posterior columns, followed by degeneration in the spinocerebellar and corticospinal tracts of the spinal cord [105,106]. Satellite cells and Schwann cells also seem to be affected in FRDA [107] but there is no clear evidence whether the observed phenotype may result from abnormal satellite cell or Schwann cell-neuron interactions. Both lack of myelination of large sensory neurons axons in the DRG and axonal degeneration have been described [108] and axonopathy has been associated with a dying-back mechanism [109]. Moreover, lesions in dentate nuclei and Purkinje cellsin cerebellum have been observed and account for the cerebellar phenotype [107]. Degeneration of enolase (NSE)-reactive neurons and of large inhibitory GABAergic terminals, as well as alterations in axonal connections of these neurons with Purkinje cells have also been described [110]. In addition to central and peripheral nervous systems degeneration, patients show primary non-neurological features as cardiomyopathy and increased incidence of diabetes mellitus [111]. Heart failure and supraventricular arrhythmias resulting from cardiomyopathy are the most common cause of death in FRDA [112].

FRDA is caused by homozygous or compound heterozygous mutations in the *FXN* gene [113,114]. Specifically, 96% of patients are homozygous for an expanded GAA trinucleotide repeat in intron 1 of the gene. Normally, there are fewer than 36 GAA repeats while those affected have between 56 and 1300 repeats. The remaining 4% of patients are compound heterozygous for a GAA expansion in one allele and a different pathogenic mutation in the other allele. More than 60 different mutations have been identified and structural modeling analysis showed that mutations in the hydrophobic core of FXN alter protein stability, while mutations occurring on surface residues affect interaction with ISC machinery and heme biosynthetic proteins [115]. In most cases, compound heterozygous patients are clinically identical to patients homozygous for the GAA expansion, but few missense mutations cause an atypical or milder clinical presentation [115,116,117,118]. Specifically, a milder phenotype is observed in patients carrying missense mutations G130V, D122Y, L106S [117,118]. G130V and D122Y mutations alter FXN interaction with the ISC machinery complex, while L106S mutation alters protein stability [115]. Even if these mutations alter FXN function in different ways, it is important to note that all occur in the amino terminal half of FXN protein. Interestingly, the length of GAA expansion correlates negatively with the expression of the encoded protein and positively with the age of onset and severity of the disease [119]. 

The first molecular consequence of the GAA expansion is the decreased transcription of the *FXN* gene that leads to the expression of lower levels of a structurally and functionally normal frataxin, up to 5–30% if compared to healthy individuals [114]. The reduction in *FXN* gene transcription in FRDA is mediated by two different mechanisms. Firstly, the expanded GAA repeat leads to both histone methylation and hypoacetylation, resulting in gene silencing [120,121]. Secondly, the GAA repeat stimulates the formation of R-loop structures, DNA-RNA hybrid, and of sticky DNA [122] that block the progress of RNA polymerase II. It is worth noting that R-loops are also known to mediate the formation of repressive chromatin and therefore the presence of the R-loop may lead to both repressive chromatin formation and to RNA-polymerase II block [123].

As previously mentioned, FXN is a mitochondrial protein involved in the biogenesis of Fe-S clusters. Even if some literature data indicate that FXN might be a multifunctional protein involved in different mitochondrial pathways, only the role of FXN in Fe-S cluster biogenesis has been convincingly and extensively proved [124]. 

Correct Fe-S cluster biogenesis in mitochondria is intimately linked to cellular iron homeostasis and failure to assemble mitochondrial Fe-S proteins upon FXN deficiency results in increased iron import and availability in mitochondria through the activation of IRP1 (iron regulatory protein 1). IRP1 binds to IRE elements in the 5’ and 3’ UTR of the mRNAs of iron metabolism-related proteins, resulting in iron import and accumulation in mitochondria [125]. Pathological mitochondrial iron accumulation has been reported in the heart, liver and spleen of FRDA patients [126,127] and in several animal models [128,129,130]. However, the presence of iron deposits in neurons has not been fully proved yet. Indeed, different studies showed no iron accumulation in the dentate nuclei and DRG of patients [105,127,131]. On the other hand, a recent study involving a higher number of FRDA subjects [132] found a significant increase in iron concentration in the dentate nucleus and red nucleus in the midbrain. In agreement with this observation, a relocation of iron from neurons to glia cells in the dentate nucleus and from neurons to satellites cells in DRGs has been observed [132,133], thus suggesting a redistribution of iron from dying neurons to satellite/Schwann cells. The generation of different animal models did not fully clarify this issue. Indeed, no mitochondrial iron accumulation has been reported in conditional knockout models with FXN reduction in DRG or cerebellum [128], but it was described in the brain of a FRDA model of *D. melanogaster* [134]. Therefore, the presence of iron deposits in the nervous system in FRDA still remains a controversial topic.

Iron could be highly toxic mainly because it mediates the catalysis of reactive oxygen species (ROS) by Fenton reaction. Formation of ROS results in glutathione depletion, lipid peroxidation and cell death [12]. Increased ROS levels have been described in different tissues of mice lacking FXN, including neurons [130]. However, other studies suggest ROS independent mechanisms at the basis of neurodegeneration. The most interesting one showed that loss of FXN in nervous system in flies and mice results in iron accumulation, increased sphingolipids synthesis, and in the activation of the PDK1/Mef2 (3-phosphoinositide dependent protein kinase-1/myocyte enhancer factor-2) pathway [135,136]. Mef2 is an important transcription factor that modulates the expression of multiple genes and its activity is regulated post-transcriptionally by the activity of different kinases, including PDK1 [137]. Mef2 participates in essential molecular events in the central nervous system, as neuronal development and survival and synaptic plasticity, by modulating the expression of genes involved in the regulation of neuronal activity as well as genes involved in energy storage and immune response [138]. Mutations in Mef2 have also been described in Rett-like disorder that is characterized by motor abnormalities and abnormal hand movements [139].

Despite several evidences showing the pathological role of iron accumulation in mitochondria, its implications in the pathophysiology of FRDA still remain elusive. Indeed, many studies suggest that iron accumulation represents only a late event in FRDA [128,140,141] and accumulation of iron has surprisingly been proposed as a protective mechanism in the liver, where iron is used to sustain defective heme synthesis and Fe-S cluster biogenesis occurring upon loss of FXN [125]. Moreover, as discussed above, the presence of iron deposits in neurons has not been fully proved yet. Therefore, more studies are needed to clarify the role of iron accumulation in the pathogenesis of FRDA neurodegeneration.

Fe-S clusters are involved in a plethora of pathways in cells and therefore abnormal FXN function can cause multiple and cumulative cellular dysregulations, independently of iron accumulation in mitochondria. Indeed, deficits in Fe-S cluster-containing enzymes, including TCA cycle enzymes (aconitase) and mitochondrial respiratory complexes I, II and III have been observed in different cellular and animal models of *FXN* deficiency. The reduced activity of these enzymes leads to changes in metabolic flow and to reduced mitochondrial ATP production. Mitochondrial dysfunctions associated with reduced ATP levels have been described in patient-derived cells [142], and in neurons [135,143,144] and other tissues [145,146] from multiple models of FRDA. Consistently, overexpression of FXN causes upregulation of TCA flux, respiration and ATP content [147]. Notably, reduction of Fe-S cluster-containing enzymes has not always been observed in FRDA models. Indeed, in patient derived DRGs samples a slight decrease of aconitase only—but not in respiratory complexes I, II and III—has been described [127]. Therefore, whether FXN deficiency always results in Fe-S cluster-containing enzymes deficits still need to be fully elucidated. Neurons are highly energy demanding cells and it is tempting to speculate that they could be specifically sensitive to cellular energy dysregulation. However, the reason why only large sensory neurons and neurons in the dentate nuclei in the cerebellum are involved still remains an intriguing unsolved question.

Another important Fe-S cluster containing enzymes is FECH, the last enzyme involved in heme biosynthesis. Deregulated heme synthesis and decreased heme levels have been observed in FXN deficient models but it was never reported in FRDA neurons [102,129,148]. Alteration of heme metabolism has been proposed as a pathogenetic event in other types of ataxias suggesting that a similar phenotype could be observed in FRDA. However more studies are needed to clarify the involvement of heme in FRDA.

Calcium homeostasis is an additional processes linked to FXN deficiency in neurons. Interestingly, altered calcium levels lead to failure of retrograde axonal transport along DRG axons in mice and flies [13]. Autophagy in neurons is highly dependent on axonal transport and a decrease in autophagic flux associated with high levels of ubiquitinylated proteins and damaged mitochondria has also been observed in DRGs [13]. Therefore, loss of FXN might result in calcium dyshomeostasis, altered axonal transport and aberrant autophagy. To note, other works suggest that FXN loss results instead in increased autophagy. However, in this case, if increased autophagy is a neurodegenerative mechanism [141] or a defense mechanism against oxidative stress [144] is yet to be clarified.

Alterations in actin cytoskeleton have been observed in FXN deficient DRGs. Specifically, loss of FXN results in altered actin dynamics leading to aberrant changes in the growth cone of neurons. Interestingly, hyperactive cofilin, that is an actin binding protein, emerges as the link between frataxin deficiency and actin cytoskeleton alterations [14].

In conclusion, different mechanisms have been proposed to be at the basis of neurodegeneration in FRDA and an overview is shown in Figure 2. However, which of them is the primary event triggering the neurological phenotype is still unclear. Due to the complexity of FRDA pathophysiology, probably more than one mechanism is involved. To note, it has been demonstrated that FXN deficiency causes tissue-specific changes in several metabolic pathways [149]. 

Analysis of FRDA patients clearly reveals a selective neurological and pathological pattern in FRDA in which specific populations of neurons are more susceptible to FXN deficiency. Indeed, neurodegeneration specifically involves large sensory neurons in the DRG, neurons of the posterior column of the spinal cord, neurons of the dentate nucleus and connections between these neurons and Purkinje cells in cerebellum. The identification of this specific susceptibility to FXN deficiency will definitely help to identify the precise molecular mechanism and to define therapeutic approaches to FRDA.

## 5. X-Linked Sideroblastic Anemia with Ataxia (XLSA/A)

Sideroblastic anemias are a group of disorders mainly characterized by hypochromic microcytic erythrocytes and mitochondrial iron accumulation in erythrocyte precursors [150]. X-linked sideroblastic anemia with ataxia (XLSA/A) is a rare form of inherited sideroblastic anemia characterized by an X-linked mode of inheritance [151]. Interestingly, in addition to anemia [6], XLSA/A patients present with motor delay and evidence of spinocerebellar dysfunction, including an early onset ataxia associated with severe cerebellar hypoplasia [61,152]. Intention tremor and dysarthria might also be present [6].

XLSA/A is due to mutations in the *ABCB7* gene [151] that encodes for an ABC transporter located in the inner membrane of mitochondria. ABC family of transporters is a large family of multiple transmembrane proteins involved in energy-dependent transport of a wide range of substrates across membranes [153]. Four distinct mutations have been reported in four different XLSA/A families. All described variants are missense mutations and occur in a region thought to be involved in binding of the transported substrate [154,155,156]. Phenotypic complementation assays in yeast showed that these mutations have mild phenotypic effects, suggesting that they are partial loss-of-function mutations in vivo [61] and that a more severe allele would be lethal in humans. Indeed, study of *Abcb7*-deficient mice showed that the protein is essential during development. Moreover, systemic and tissue-specific deletion of *Abcb7* in most organs, including central nervous system and bone marrow, was lethal, with the exception of liver and endothelial cells [60]. 

The molecular mechanisms linking *ABCB7* mutations to ataxia are poorly understood. As mentioned before, ABCB7 transports a sulfur-containing compound (X-S) from mitochondria to the cytosol used to generate the Fe/S clusters for cytosolic and nuclear target proteins as well as for those associated to the endoplasmic reticulum. Recent studies suggested that a glutathione-coordinated [2Fe-2S] cluster is the substrate of the transporter [157]. Indeed, the analysis of different animal and cellular models showed a selective deficiency of cytosolic, but not mitochondrial, Fe-S proteins upon ABCB7 downregulation [158]. Reduction of cytosolic Fe-S proteins may result in the alteration of a plethora of cellular mechanisms as altered DNA repair, glycolysis, fatty acid synthesis, purine catabolism and ribosome biogenesis [159]. To note, loss of *ABCB7* in HeLa cells reduces the activity of aconitase [160], suggesting that the alteration of energetic metabolism may play a role in the disease pathogenesis.

An additional mechanism that may contribute to the ataxic phenotype in XLSA/A is represented by defective heme synthesis. Indeed, loss of ABCB7 alters the activity of IRP1, which is responsive to the availability of cytosolic Fe-S clusters. This results in the inhibition of the expression of ALAS2 that is involved in heme synthesis, impaired cellular iron homeostasis and mitochondrial iron overload [60,161]. The accumulated mitochondrial iron has been shown not to be available to mitochondrial ferritin and so, it is not readily usable for heme synthesis [160]. Moreover, the interaction between ABCB7 and FECH, the last enzyme in heme biosynthesis, has been demonstrated and ABCB7 positively regulates the activity of this enzyme [31]. Therefore, loss of ABCB7 directly or indirectly inhibits heme synthesis. 

Although the majority of these studies have been performed in erythroid cells, it is likely that defects in energetic metabolism and/or heme synthesis may contribute to the degeneration of neural circuity regulating balance and coordination of movement as well.

## 6. Therapeutic Approaches

There are no therapeutic options that actually halt the progression of PCARP, FRDA and XLSA/A. Therefore, the management of these patients is mainly symptomatic. In particular, FRDA has a complex and variable clinical phenotype and therefore it requires a broad multidisciplinary approach to manage [12]. 

Physiotherapy provides an important tool for the maintenance of balance, flexibility, strength and accuracy of limb movements, all of which can help to improve the functional consequences of gait and limb ataxia. Occupational therapy is also important and allows for the assessment and optimization of functional status, thereby reducing the impediments to activities of daily living.

In FRDA, the heart requires special attention because hearth failure is a common cause of mortality and therefore, electrocardiogram and echocardiography should be performed at diagnosis. Many patients show impaired glucose tolerance or diabetes and should be counseled on the importance of dietary changes and physical exercise [12].

Progresses in the identification of novel strategies to cure or limit the disease progression has been made only for FRDA. This is likely due to a better understanding of the molecular pathogenesis and the higher frequency of the diseases allowing the design of clinical trials. A number of molecules targeting specific pathological processes have been evaluated in clinical trials. Earlier clinical trials attempted to resolve oxidative stress and iron accumulation by using antioxidant and iron chelation agents, both as monotherapy and in combination, but failed to give consistent results [162,163]. However, recently the antioxidant drug omaveloxolone (RTA408), an activator of the anti-oxidant nuclear factor, erythroid 2 like 2 protein (Nrf2), has shown promising results as a treatment for FRDA [164]. A Phase II study is currently evaluating the safety, pharmacodynamics and efficacy of omaveloxolone [162].

To date, heterozygous carriers with almost 50% level of *FXN* transcript remain asymptomatic [123]. This observation indicates that even a small increase in *FXN* transcript levels is likely to be therapeutic. Therefore, another possible approach is treating FRDA by restoring FXN levels. For example, erythropoietin has been shown to increase *FXN* levels in FRDA models, but clinical trials failed to demonstrate clinical efficacy of this molecule [165,166]. In general, these compounds produce small increases in *FXN* protein levels. A greater increase can be achieved by gene therapy. Gene therapy requires the administration of adeno-associated virus (AAV) carrying human FXN. Their administration in two different mouse models of FRDA, both before and after the insurgence of heart failure or neurological dysfunctions, was sufficient to inhibit or complete revers cardiomyopathy or sensory neuropathy, respectively [167,168]. Different clinical trials are now underway for the treatment of FRDA using gene therapy. However, some immunological complications are present. Interestingly, synthetic lipid nanoparticles have been used to deliver *FXN* mRNA and this approach has the potential to overcome some of the immunological limitations associated with viral delivery systems [162,169].

More recent approaches focused on epigenetic elements. For example, inhibition of histone deacetylase (HDAC) has emerged as a promising therapeutic strategy [170]. An alternative strategy uses oligonucleotides to disrupt R-loop formation, and also showed partial reversal of *FXN* gene silencing [171]. Similarly, dimethyl fumarate reduces the formation of R-loop structures and increases transcript of *FXN* [172]. Moreover, syn-TEF1, a synthetic transcription factor, specifically reversed the block of polymerase II and showed partial reactivation *FXN* gene transcription [173].

Finally, thanks to recent advances in genome editing, deleting, or at least reducing the expanded GAA repeat might represent another possible therapeutic approach.

## 7. Concluding Remarks

In the present review, we discussed current knowledge on the molecular mechanisms underlying PCARP, FRDA and XLSA/A. Although the three forms of ataxia are clinically heterogeneous, they are undoubtedly caused by mutations in genes involved in strongly interconnected pathways. As already discussed, heme synthesis depends on Fe-S cluster availability and we cannot exclude that heme metabolism may influence Fe-S cluster levels. 

Strong progresses in the understanding of the molecular pathogenesis of FRDA ataxia has been made in recent years, whereas less is known about PCARP and XLSA/A. Due to the involvement of heme and Fe-S clusters in multiple essential processes, it is conceivable that multiple mechanisms could be active in these disorders. However, the observation that heme and Fe-S clusters regulate common pathways, like energetic cell metabolism, suggest the possibility that similar pathogenetic mechanisms may be responsible for XLSA/A and PCARP. Eventually, the identification of common pathways altered in these forms of ataxia could be important for drug repurposing strategies. Future research should be directed to a better elucidation of the molecular mechanisms underlying these rare diseases and to the identification of novel therapeutic options. 

## Figures and Tables

**Figure 1 ijms-21-03760-f001:**
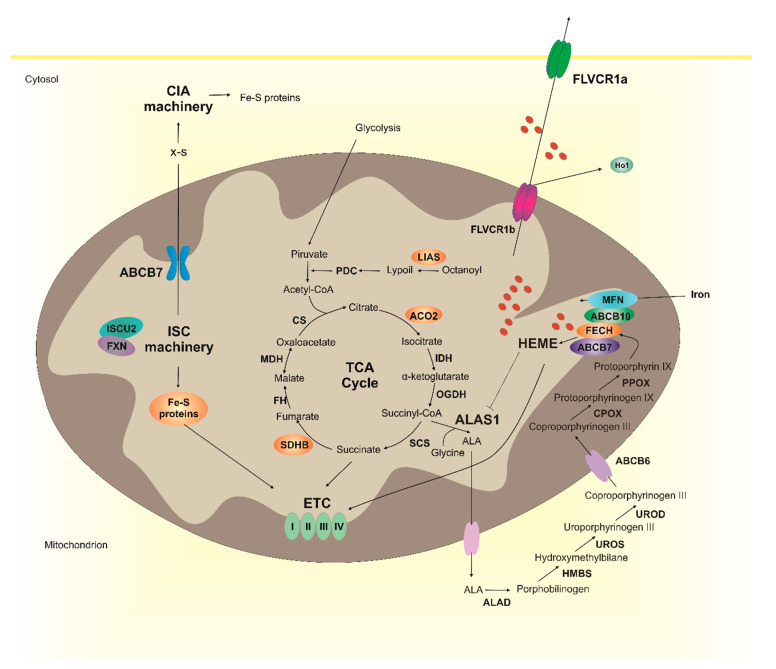
Heme metabolism and Fe-S cluster biogenesis are interconnected. The ISC machinery (composed by ISCU2, FXN, and other 16 proteins) is responsible for the generation of Fe-S clusters necessary for several mitochondrial proteins (in orange), including those of the electron transport chain (ETC), some enzymes of the TCA cycle (ACO2 and SDHB) and LIAS. The ISC machinery is also responsible for the generation of a X-S compound that is exported in the cytosol by ABCB7 and used by the CIA machinery for the generation of Fe-S clusters to be incorporated into non-mitochondrial proteins. Heme is synthesized through eight enzymatic reactions occurring between mitochondria and cytosol. The availability of free heme is further controlled at the level of heme catabolism (HO1) and heme export (FLVCR1a and FLVCR1b). FLVCR1b has been drawn on both mitochondrial membranes, however its specific subcellular localization still remains to be determined. Although not shown in the picture other heme importers and exporters have been identified (see the text for details). Both heme and Fe-S clusters are necessary for the ETC activity. Interestingly, the biosynthetic pathways of the two cofactors are strongly interconnected. Indeed, heme synthesis depends on Fe-S clusters for succinyl-CoA supply, for the expression of ALAS1 and FECH. Whether Fe-S clusters biogenesis depends on heme is less clear.

**Figure 2 ijms-21-03760-f002:**
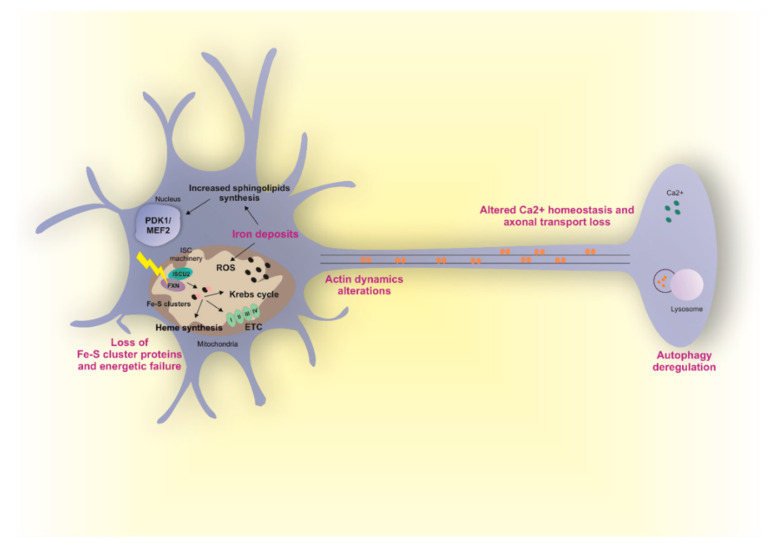
Altered molecular mechanisms in FRDA affected neurons. Frataxin (FXN) deficiency in FRDA results in the degeneration of specific neuron types leading to the ataxia phenotype. Different hypotheses have been postulated regarding the pathophysiological mechanism at the basis of neurodegeneration. FXN is a mitochondrial protein involved in Fe-S clusters biogenesis and alterations in FXN activity results in mitochondria dysfunctions and iron accumulation. The presence of iron deposits in mitochondria has been observed in patient-derived samples; however, whether iron accumulation occurs in neurons is less clear. Iron might be toxic and cause cellular dysfunctions by mediating the catalysis of ROS via Fenton reaction. However, a ROS-independent mechanism of iron toxicity has also been proposed. In this model, neurodegeneration arises from increased sphingolipids synthesis and activation of PDK1/Mef2 pathway occurring upon iron accumulation. Independently of iron accumulation, FXN deficiency results in reduced activity of Fe-S cluster containing enzymes leading to alteration in metabolic flow (aconitase), energetic failure (ETC), and altered heme synthesis (FECH). Moreover, a role of FXN in the regulation of calcium homeostasis has been described and linked to failure of retrograde axonal transport along axons and autophagy deregulation. Aberrant autophagy levels have been also reported in other independent works. However, whether autophagy is a neurodegenerative mechanism or a defense mechanism still need to be clarified. Finally, FXN deficiency may control actin dynamics and the growth cone of neurons. All hypotheses have been well described. However, which of them is the primary event triggering the neurological phenotype in FRDA is still unknown.

**Table 1 ijms-21-03760-t001:** Main clinical features and sites of pathology of PCARP, FRDA and XLSA/A.

	Gene	Function	Neurological Clinical Features	Sites of Pathology	Ref.
*PCARP*	*FLVCR1*	Heme export	Sensory ataxia Retinitis pigmentosa	Posterior columns of spinal cord Retina	[8,9]
*FRDA*	*FXN*	Fe-S cluster biogenesis	Mixed spinocerebellar and sensory ataxiaDysarthriaMuscular weakness Absence of tendon reflexes with deep sensory loss	Large sensory neuronsPosterior columns of spinal cordDentate nucleus of cerebellum	[10,11,12]
*XLSA/A*	*ABCB7*	Export of S-X compound from mitochondria	Spinocerebellar ataxia Intention tremor Dysarthria	Cerebellum	[13,14]

**Table 2 ijms-21-03760-t002:** *FLVCR1* mutations reported in PCARP patients so far.

Mutation(s)	Zygosity	Types of Mutation	Exon/Intron	Ref.
c.361A>G (p.Asn121Asp)	Homozygous	Missense	Exon 1	[79,81]
c.721G>A (p.Ala241Thr)	Homozygous	Missense	Exon 1	[79]
c.574T>C (p.Cys192Arg)	Homozygous	Missense	Exon 1	[79]
c.1477G>C (Gly493Arg)	Homozygous	Missense	Exon 8	[83]
c.1547G>A (p.Arg516Gln) c.1593+5 +8delGTAA	Compound Heterozygous	Missense Splicing	Exon 9 Intron 9–10	[82]
c.596T>C (p.Leu199Pro)	Homozygous	Missense	Exon 1	[80]

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
