# Peer review of "Hereditary Ataxia: A Focus on Heme Metabolism and Fe-S Cluster Biogenesis"

_ijms, 2020, doi:10.3390/ijms21113760_

Round 1

Reviewer 1 Report

In this manuscript Chiabrando and collaborators discuss recent progresses in the understanding of the molecular pathogenesis of Posterior Column Ataxia and Retinitis Pigmentosa (PCARP), Friedreich’s ataxia (FRDA) and X-13 linked sideroblastic anaemia with ataxia (XLSA/A). These diseases are caused by mutations in genes involved in heme metabolism and Fe-S cluster biogénesis.

The manuscript presents a comprehensive review on the subject. However, there are some points that should be addressed:

 - In Figure 1, Frataxin is indicated FTX. FTX is also used in other parts of the text (for instance lines 438-440). Please use always the same abbreviation (FXN or FTX)

- In Figure 1, line from heme to ETC is discontinous, while that from FeS is continuous. As both FeS and heme are required for ETC activity, both lines should be equal.

- Some metabolites in Figure 1 are hard to read. Bigger size letter should be used. For clarity, NAD/NADH could be avoided (actually not all reactions generating it have been indicated, for instance PDC).  

 - The statements in line 326 and line 333 are contradictory: “Indeed, absence of FXN causes generalized Fe-S cluster-containing enzyme deficiency”, while on the following paragraph “However, in patient derived DRGs samples a slight decrease of aconitase only, but not in respiratory complexes I, II and III, has been observed”. Actually, FeS deficiency is not always observed in frataxin-deficient cells. Therefore on line 326 authors should avoid giving the idea that frataxin deficiency always “causes generalized FeS deficiency”, because this  is not true. Indeed, authors should place more emphasis on the evidence that FeS deficiency is not always observed in frataxin deficient cells.  

 - Line 362. “To note, it has been recently demonstrated that FXN deficiency causes tissue-specific changes in several metabolic pathways (147).” Ref 147 is from year 2009. It can not be considered a “recently” observation.

Author Response

In this manuscript Chiabrando and collaborators discuss recent progresses in the understanding of the molecular pathogenesis of Posterior Column Ataxia and Retinitis Pigmentosa (PCARP), Friedreich’s ataxia (FRDA) and X-13 linked sideroblastic anaemia with ataxia (XLSA/A). These diseases are caused by mutations in genes involved in heme metabolism and Fe-S cluster biogénesis. The manuscript presents a comprehensive review on the subject.

We thank the reviewer for the positive evaluation of the manuscript.

However, there are some points that should be addressed:

 - In Figure 1, Frataxin is indicated FTX. FTX is also used in other parts of the text (for instance lines 438-440). Please use always the same abbreviation (FXN or FTX).

Thank you for the observation. We used the abbreviation FXN both in the figure and in the text.

- In Figure 1, line from heme to ETC is discontinous, while that from FeS is continuous. As both FeS and heme are required for ETC activity, both lines should be equal.

Thank you for the suggestion. We modified the line from heme to ETC and now it appears as a continuous line.

- Some metabolites in Figure 1 are hard to read. Bigger size letter should be used. For clarity, NAD/NADH could be avoided (actually not all reactions generating it have been indicated, for instance PDC).

Thank you for the suggestions. We changed the character size and eliminate NAD/NADH ratio from the figure.

 - The statements in line 326 and line 333 are contradictory: “Indeed, absence of FXN causes generalized Fe-S cluster-containing enzyme deficiency”, while on the following paragraph “However, in patient derived DRGs samples a slight decrease of aconitase only, but not in respiratory complexes I, II and III, has been observed”. Actually, FeS deficiency is not always observed in frataxin-deficient cells. Therefore, on line 326 authors should avoid giving the idea that frataxin deficiency always “causes generalized FeS deficiency”, because this is not true. Indeed, authors should place more emphasis on the evidence that FeS deficiency is not always observed in frataxin deficient cells.

We thank the reviewer for the comment. We modified the sentences in order to better explain this issue (Lanes 342-345).

 - Line 362. “To note, it has been recently demonstrated that FXN deficiency causes tissue-specific changes in several metabolic pathways (147).” Ref 147 is from year 2009. It can not be considered a “recently” observation.

We removed the word “recently” from the text.

Reviewer 2 Report

In this manuscript, the authors reviewed the biology of heme metabolism and Fe-S cluster biogenesis, and its relevance in the pathogenesis of three forms of ataxia [Posterior Column Ataxia and Retinitis Pigmentosa (PCARP), Friedreich’s ataxia (FRDA) and X-14 linked sideroblastic anaemia with ataxia (XLSA/A)]. Recent progresses in therapeutic options for these pathologies, particularly for FRDA, are also discussed. In overall, the manuscript is well written, provides pertinent insights about the relevance of heme and Fe-S clusters in the pathogenesis of ataxias, and gives an interesting overview about a topic that has not been clearly clarified.

I have some minor comments:

  1. Line 15: “…molecular pathogenesis of these disorders several important questions…” – a comma is missing - “…molecular pathogenesis of these disorders, several important questions…”

  1. Line 29: Actually, there are more than 70 types of hereditary ataxia described. Can the authors add a more recent reference?

  1. Line 82: “…target apoproteins that resides in different subcellular…” – minor typo -”…target apoproteins that reside in different subcellular…”

  1. Line 96: “…as heme exporter.” – minor typo - “…as heme exporters.”

  1. Line 100: “Consistently, heme export resulted co-regulated with heme synthesis.” The meaning of the sentence is not clear, please clarify.

  1. Line 158: “…large complex of proteins, included” – minor typo “…large complex of proteins, including”

  1. Line 134 (Figure 1): In the ISC machinery, FTX should be FXN? The same in lines 438 and 440.

  1. Line 262: “…but few missense mutations cause an atypical or milder clinical phenotype.” The missense mutations that the authors mentioned are the ones located at the N-terminal of the protein? Because there seems to be a difference in the phenotype caused by missense mutations at the C-terminal or at the N-terminal. Can the authors clarify this issue?

  1. There is no reference in the text to figure 2, please add it.

Author Response

In this manuscript, the authors reviewed the biology of heme metabolism and Fe-S cluster biogenesis, and its relevance in the pathogenesis of three forms of ataxia [Posterior Column Ataxia and Retinitis Pigmentosa (PCARP), Friedreich’s ataxia (FRDA) and X-14 linked sideroblastic anaemia with ataxia (XLSA/A)]. Recent progresses in therapeutic options for these pathologies, particularly for FRDA, are also discussed. In overall, the manuscript is well written, provides pertinent insights about the relevance of heme and Fe-S clusters in the pathogenesis of ataxias, and gives an interesting overview about a topic that has not been clearly clarified.

We thank the reviewer for the positive evaluation of the manuscript.

I have some minor comments:

  1. Line 15: “…molecular pathogenesis of these disorders several important questions…” – a comma is missing - “…molecular pathogenesis of these disorders, several important questions…”

We inserted the comma.

  1. 29: Actually, there are more than 70 types of hereditary ataxia described. Can the authors add a more recent reference?

More recent references were added. Even if one review was written in 1993, it was updated in 2019 and shows different tables with all different types of ataxia.

  1. Line 82: “…target apoproteins that resides in different subcellular…” – minor typo -”…target apoproteins that reside in different subcellular…”

We corrected the sentence.

  1. Line 96: “…as heme exporter.” – minor typo - “…as heme exporters.”

We corrected the sentence.

  1. Line 100: “Consistently, heme export resulted co-regulated with heme synthesis.” The meaning of the sentence is not clear, please clarify.

We thank the reviewer for the comment that allow us to better clarify the issue. Literature data indicate that the expression of the heme exporter FLVCR1 and the rate limiting enzyme in heme biosynthesis ALAS1 are co-regulated (up-regulated) following the stimulation of heme synthesis, at least in hepatocytes. We modified the text in order to better explain this point (lanes 100-101).

  1. Line 158: “…large complex of proteins, included” – minor typo “…large complex of proteins, including”

We corrected the sentence.

  1. Line 134 (Figure 1): In the ISC machinery, FTX should be FXN? The same in lines 438 and 440.

We corrected the abbreviations.

  1. Line 262: “…but few missense mutations cause an atypical or milder clinical phenotype.” The missense mutations that the authors mentioned are the ones located at the N-terminal of the protein? Because there seems to be a difference in the phenotype caused by missense mutations at the C-terminal or at the N-terminal. Can the authors clarify this issue?

We thank the reviewer for the comment. With milder clinical phenotype we referred to mutations: G130V, D122Y, L106S. R165P has been associated with an atypical but not milder phenotype. G130V and R165P mutations alter FXN interaction with ISCU, while D122Y alters the interaction with ferrochelatase and ISCS. Differently, L106S mutation alters protein stability.

Even if G130V, D122Y, L106S mutations alter FXN function in different ways, they occur in exon 3 and 4, so in the amino terminal part of FXN protein, thus suggesting that missense mutations occurring in the amino terminal half result in a milder phenotype.

We modified the text in order to better clarify this issue (lanes 264-273).

  1. There is no reference in the text to figure 2, please add it.

We added the reference (line 370).